# Culturally adapted pulmonary rehabilitation for adults living with post-tuberculosis lung disease in Kyrgyzstan: protocol for a randomised controlled trial with blinded outcome measures

Azamat Akylbekov [1,2] Mark W Orme [3,4] Amy V Jones [3,4]
Maamed Mademilov,[1,2] Aibermet Muratbekova,[1,2] Shoira Aidaralieva,[1,2]
Gulzada Mirzalieva,[1,2] Alena Oleinik,[1,2] Kamila Magdieva,[1,2] Aijan Taalaibekova,[1,2]
Aidai Rysbek kyzy,[1] Zainab K Yusuf [3,4] Jones Rupert,[5] Andy Barton,[6]
Ruhme B Miah,[3] Adrian Manise,[7] Jesse A Matheson,[8] Dominic Malcolm,[9]
Robert C Free,[3] Michael C Steiner,[3,4] Talant Sooronbaev,[1,2] Sally J Singh[3,4]

For numbered affiliations see end of article.

**Correspondence to**
Dr Azamat Akylbekov;
azamatti@yahoo.com

## ABSTRACT

**Introduction** Pulmonary rehabilitation (PR) is a programme of individually prescribed physical exercise, education and self-management activities. PR is recommended in international guidelines for managing chronic obstructive pulmonary disease (COPD) and other chronic respiratory diseases. PR is still under-recognised in tuberculosis (TB) guidelines and PR is not available in many low and middle-income countries and for people with post-TB lung disease (PTBLD). The main aims of the study are to adapt and define a culturally appropriate PR programme in Kyrgyzstan for people living with PTBLD and to test, in a fully powered randomised controlled trial (RCT), the effectiveness of PR in improving exercise capacity for people living with PTBLD.

**Methods and analysis** The study will be divided into three stages: *stage 1:* focus group discussions with patients living with PTBLD and interviews with PR referrers will be conducted to explore initial perceptions and inform the cultural adaptation, structure and content of PR. *Stage 2a:* a single-blind RCT evaluating the effectiveness of a culturally adapted 6-week PR programme on maximal exercise capacity, assessed by the incremental shuttle walking test, before and after PR. Participants will be additionally followed-up 12 weeks postbaseline. Additional outcomes will include health-related quality of life, respiratory symptoms, psychological well-being and physical function. *Stage 2b:* participants' experience of PR will be collected through interviews and using a log book and a patient evaluation form. Staff delivering PR will be interviewed to explore their experience of delivering the intervention and refining the delivery for future implementation.

**Ethics and dissemination** The study was approved 22/07/2019 by Ethics Committee National Center for Cardiology and Internal Medicine (reference number 17) and by University of Leicester ethics committee (reference number 22293). Study results will be disseminated through appropriate peer-reviewed journals, national and international respiratory/physiotherapy conferences, social media, and through patient and public involvement events in Kyrgyzstan and in the UK.

**Trial registration number** ISRCTN11122503.

## Strengths and limitations of this study

► This project will inform an appropriate pulmonary rehabilitation programme for people living with post-tuberculosis lung disease (PTBLD) in Kyrgyzstan. This study is the first fully powered randomised controlled trial (RCT) of a culturally appropriate pulmonary rehabilitation programme for adults with PTBLD in Central Asia.

► This is a single-centre fully powered RCT; while findings will be important for assessing impact of pulmonary rehabilitation in Kyrgyzstan, Central Asia and low and middle-income countries (LMIC) more broadly, findings may not be generalisable to other parts of the world.

► Taking a mixed-method approach, this study will provide a rich insight into the experiences of pulmonary rehabilitation for people living with PTBLD.

► Using a common set of outcome measures specifically designed for pulmonary rehabilitation in LMIC will allow for direct comparisons of outcomes between sites across the world.

## INTRODUCTION
### Background and rationale

Tuberculosis (TB) is an infectious disease and a major cause of death worldwide.[1] It typically affects the lungs (pulmonary TB) and approximately a quarter of people globally are at risk of developing TB.[1] Every year, approximately 10 million people worldwide develop TB,[2] of which 500000 are multidrug-resistant cases.[3] Pulmonary TB remains an important cause of chronic respiratory impairment in low and middle-income countries (LMIC) and globally. Kyrgyzstan is classified as lower middle income country, is situated in Central Asia and has a higher respiratory mortality compared with Europe, with a TB incidence of 100:100000.[4]

The presence of long-term respiratory sequelae following pulmonary TB treatment is well established.[5 6] The persistence of abnormal airway physiology,[7] specifically fixed airways obstruction after treatment, has been documented in large population-based cross-sectional studies. The consequence of suffering from pulmonary TB is termed post-TB lung disease (PTBLD). PTBLD negatively impacts people's quality of life, with exercise intolerance, respiratory symptoms and abnormal lung function.[8 9] PTBLD also poses a significant economic burden to individuals and societies, not only attributable to direct costs of drugs and hospital stay but also indirect costs such as missed workdays or lower productivity. The burden of PTBLD alongside the economic conditions in LMIC such as Kyrgyzstan aggravates the situation of patients, leading to disability and early mortality.[10]

Pulmonary rehabilitation (PR) is a programme of individually prescribed physical exercise, education and self-management activities. PR is recommended in international guidelines for managing chronic obstructive pulmonary disease (COPD) and other lung diseases[11] but is still underrecognised in TB guidelines and in LMIC.[12] With the established benefits of PR for people living with chronic respiratory diseases, many high-income countries have integrated PR services within routine healthcare services. Delivery of PR is encouraged in LMIC where there is profound need,[13] where there is emerging evidence that it is feasible to deliver PR in low-resource settings,[14 15] including for people living with PTBLD[16] for whom post-treatment support for health and well-being is needed.[17] Indeed, PR is typically delivered by a multidisciplinary team, which has been advocated for people living with PTBLD in order to account for psychosocial and economical challenges.[18 19] Western models of PR may not be optimal in LMIC and there is a need to implement PR interventions tailored to the local culture, traditions, geography, population and healthcare systems. To date, despite the potential to improve patient outcomes in PTBLD, there has been no formal fully powered trial of PR for people living with PTBLD in LMIC. It is important for PR to be appealing to patients and to ensure that it is delivered in a manner sensitive to local contexts. Accordingly, the aim of this study is to adapt a conventional model of PR to Kyrgyzstan and PTBLD contexts and test its effectiveness in a fully powered randomised controlled trial (RCT).

The trial is comprised of: stage 1: adaptations to PR informed by patients living with PTBLD and healthcare professionals (HCPs) who would refer to PR; stage 2a: fully powered RCT; and stage 2b: qualitative evaluation of PR. The objectives of this study are to:

1. Explore the views of people living with PTBLD and HCPs (referrers and deliverers) involved in the care of patients living with PTBLD to inform the adaptations needed for a PR programme suitable for people living with PTBLD in Kyrgyzstan.
2. Conduct a single-blind fully powered RCT to assess the effectiveness of adapted 6-week PR on maximal exercise capacity (assessed by the incremental shuttle walking test (ISWT)).
3. Evaluate a range of secondary outcome measures including health-related quality of life, respiratory symptoms, functional status and psychological well-being following PR.
4. Assess any further changes in all outcome measurements 12 weeks postbaseline.
5. Assess the acceptability of PR of participants and the staff delivering PR to inform further improvements to the service.

## METHODS AND ANALYSIS
### Trial design

The trial will be conducted, analysed and reported according to the Standard Protocol Items: Recommendations for Interventional Trials statement and the trial has been prospectively registered.

### Stage 1

Focus groups with adults living with PTBLD and interviews with staff who would be involved in referring patients to PR will be conducted to explore initial perceptions of PR and to inform the cultural adaptation of PR, for their insight, any minor modifications and any additional needs, so that it is suitable for this group and for understand their views on any specific topics for the health education component, or any types of exercises that are suitable for enjoyment and adherence for this population.

### Stage 2A

After the PR programme has been adapted based on the qualitative work at stage 1, an RCT evaluating the effectiveness of PR in adults with PTBLD. Adults living with PTBLD will be randomly allocated into one of two groups: group A (assigned PR programme) or group B (usual care on PR waiting list).

### Stage 2B

On completion of PR, patient focus groups and interviews with staff who were involved in the delivery of PR will be conducted. Participants in the PR group will be asked to log their experience as they progress through the programme by completing a log book accessible before, during and after sessions as well as a dedicated patient evaluation form to be completed after their last class.

## Study setting

The study will be conducted at the National Center for Cardiology and Internal Medicine (NCCIM), Bishkek, Kyrgyzstan. Interviews will be conducted in quiet and familiar settings, for example, a HCP's office. Spacious and quiet rooms will be used for focus groups. For the PR programme, a specially allocated room with the necessary equipment and air conditioner will be used. For all stages of the project, appropriate precautions will be taken in line with local COVID-19 guidance.

## Participants

For stage 1, eligible patients will be: aged ≥18 years; confirmed diagnosis of TB-negative patients with PTBLD using a Ziehl-Nielsen stain or GeneExpert method and completed TB treatment. Eligible HCPs will be directly involved in the long-term care of patients with PTBLD.

For stage 2a, the inclusion criteria for patients will be: aged ≥18 years; confirmed diagnosis of a TB-negative patients with PTBLD using a Ziehl-Nielsen stain or GeneExpert method; completed TB treatment; Medical Research Council (MRC) dyspnoea score grade 2 or higher. Exclusion criteria for patients will be: comorbidities such as severe or unstable cardiovascular, other internal diseases and locomotor difficulties that preclude exercise; malignant disease such as lung cancer; evidence of active TB on chest X-ray or sputum tests within 1 month of assessment; unable or unwilling to provide informed consent.

For stage 2b, all patients consented and randomised to receive PR and all staff involved in the delivery of PR to patients as part of the trial will be eligible.

## Procedure

After receiving a research fact sheet, patients eligible for the study will be asked if they would like to participate after they have had the opportunity to ask questions. If they wish to take part, they will be asked to provide written informed consent (Appendix A). Reasons for declining the study will be taken as field notes. If they wish to participate, they will be asked to provide written informed consent. Reasons for declining to participate will be reflected in the field notes. The staff of the family medicine centres, doctors of the City Tuberculosis Hospital, the City Tuberculosis Dispensary, the National Hospital, District Hospitals and the National Centre of Cardiology and Internal Medicine will screen patients for eligibility and refer interested patients to the PR study team. Interested patients will be screened by the study team to confirm interest in participation and to schedule the baseline visit. After providing written informed consent, participants will undergo a baseline assessment at the NCCIM by a specially trained team of researchers and residents. After the baseline measures are complete, participants will be randomised into one of two groups: PR or usual care control. At 6-week postbaseline and 12-week postbaseline, patients will undergo follow-up assessments. If participants are not able to attend

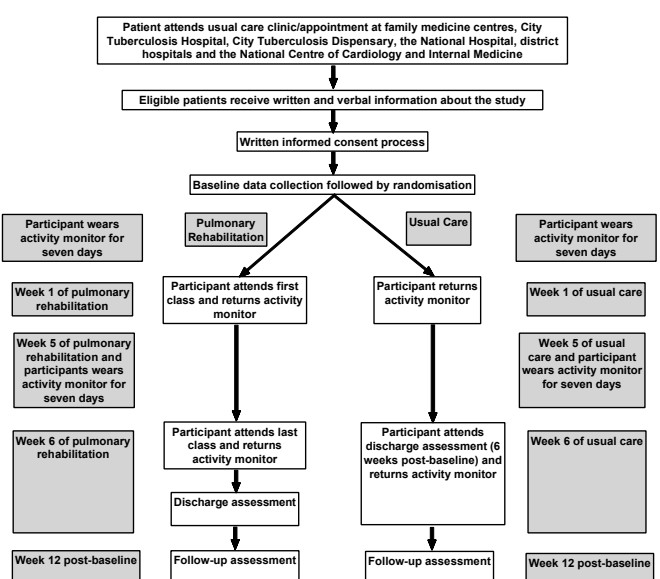

**Figure 1** Research flow. The stages and flow of research are clearly described here.

follow-up assessments in person, questionnaires only will be completed via telephone appointment. Experiences of the patients living with PTBLD and PR deliverers will be explored in interviews and focus groups. Participants who drop out of the trial will be asked to take part in an interview and information provided freely by the participants will be collected as field notes. All outcome measures will be performed at 6 and 12 weeks by dedicated staff who are blinded to group allocations. Control participants will be offered PR at the end of the study (see figure 1).

## Intervention

The PTBLD PR programme will be held two times weekly for a period of 6 weeks. Each session will last approximately 2 hours, consisting of exercise and health education (online supplemental table 1). The PR programme recruitment will be rolling rather than fixed groups, with a maximum capacity of five patients per session (due to pandemic) and supervised by a nurse, doctor, researcher or physiotherapist. The equipment used will be simple and based on local availability. It will be based on the core elements of an evidence-based rehabilitation, including exercises and health education, but we will be adapting the detail and delivery based on the qualitative work in Stage 1.

Core components of PR exercise are upper and lower body resistance training and aerobic training will be informed by international guidelines.[11 20] The strength exercises will include (at least) bicep curls, pull-ups, sit-to-stands and step ups. Patients will be asked to do one session of strength training and to walk every day at home in addition to structured classes. Minimal equipment will be used where possible, such as bottles filled with water, to achieve the desired weight in kilograms and to offer insight into how participants can exercise effectively at home during and after finishing PR. Aerobic exercise will include walking and static cycling. Walking will

be individually prescribed at a speed equivalent to 85% maximal oxygen capacity ($VO_2$) peak derived from the ISWT.[21 22] Walking will be monitored and target duration increased by the instructor as the programme progresses. The content delivered within the education sessions will be based on existing materials, with refinement from stage 1 of this trial and will be delivered by an interdisciplinary team.

### Usual care
As part of usual care, participants will receive their usual prescription medications (if appropriate), and an educational booklet regarding PTBLD and the importance of exercise, healthy diet, smoking cessation and avoiding biomass smoke.

### Sample size
#### Stage 1
We will conduct 8–10 focus groups of 2–10 patients (ideally 6–8 patients) in each group and up to 15 interviews with referrers to explore initial perceptions of PR and to inform the cultural adaptation of PR, for their insight, any minor modifications and any additional needs, so that it is suitable for this group and for understand their views on any specific topics for the health education component or any types of exercises that are suitable for enjoyment and adherence for this population.

#### Stage 2a
Assuming a statistical significance level of 5% and a statistical power of 80%, in order to detect a 35 m difference in ISWT measured at baseline and after completion of PR,[23] a total of 114 participants are required (PR: 57 participants or waiting list control: 57 participants). This power calculation was performed using a paired t test and assumes the ISWT follows an approximately normal distribution. Data from a feasibility study in Uganda, assessing PR in a PTBLD population, have guided this calculation.[14] Conservatively assuming up to 30% loss to follow-up at 6 weeks, this will require up to 114 participants to be recruited and randomised (1:1) to each arm. The estimated 70% ineligibility rate suggests approximately 543 patients with PTBLD will be screened for this trial.

#### Stage 2b
We will conduct up to five focus groups of 2–10 patients (ideally 6–8 patients) and an interview with all members of the PR delivery team. All participants randomised to the PR group will be asked to complete the log book and evaluation form.

### Randomisation
Randomisation will occur through a web-based service Sealed Envelope and conducted in a designated room by a dedicated member of the research team. Participants will be randomised (1:1) to either PR or usual care. A dedicated member of the research team will inform the assessors of group allocations via telephone. Participants

will be informed at the end of their baseline visit to arrange their first PR class.

### Blinding
Outcome measures taken at baseline, post-PR and follow-up will be taken by a researcher blinded to group allocation. It is not possible to blind participants to treatment due to the nature of a PR intervention. Research staff will be blinded to outcome measures. Participants will be advised not to reveal their group during the follow-up assessments. Episodes of un-blinding will be documented and reported for both time points.

### Qualitative data collection
All interview and focus group discussions will be audio-recorded, expected to last on average 20–40 min and will be conducted by a trained interviewer. Recordings will be transcribed verbatim, with identifiable information removed. Consent will be obtained from participants prior to their involvement. PR deliverers will be invited to participate in in-depth interviews at the end of the study to discuss aspects of PR, such as insights into barriers to attendance, logistical barriers of running a PR programme and their views of patients' experiences of the intervention.

### Book of testimonies and evaluation form
Participants within PR will be asked to log their experience of PR as they progress through the programme. This will be in the form of a PR log book accessible to participants before, during and after sessions as well as a dedicated evaluation form (Appendix B). Staff involved in PR will also receive the same evaluation form at the end of the study.

### Quantitative data collection
Baseline characteristics will comprise patient sociodemographics (age, sex), age of leaving full-time education, anthropometrics (height, weight, body mass index), smoking status and pack years, biomass fuel exposure, primary respiratory diagnosis, secondary respiratory diagnosis (if appropriate), spirometry data (postbronchodilation forced expiratory volume in 1 s ($FEV_1$), post-bronchodilation Forced Vital Capacity (FVC) and $FEV_1$/FVC ratio) (EasyOne Plus spirometer, NDD Medical Technologies, Switzerland); number of hospitalisations in the last 12 months, comorbidities, treatment and previous participation in PR. The height will be measured by stadiometer and the weight will be measured by electronic scales.

### Outcomes
A schedule of events is provided in table 1 and measures are in accordance with the recommended minimum data set for PR in LMIC.[16] In the case of a face-to-face follow-up, appointment at 12-week postbaseline cannot be conducted, questionnaires will be collected over telephone.

**Table 1** Schedule of events

| | Baseline | 6 weeks | 12 weeks | |
| --- | --- | --- | --- | --- |
| | Face-to-face | Face-to-face | Face-to-face | Telephone* |
| Sociodemographics | x | | | |
| Age of leaving full time education | x | | | |
| Smoking and biomass exposure | x | | | |
| Respiratory diagnoses | x | | | |
| Spirometry | x | | | |
| Hospitalisations in the last 12 months | x | | | |
| Comorbidities | x | | | |
| Treatments | x | | | |
| Physical activity monitoring | x | x | | |
| Anthropometrics | x | x | x | |
| ISWT and ESWT | x | x | x | |
| 5x sit to stand | x | x | x | |
| MRC score | x | x | x | x |
| EQ5D score | x | x | x | x |
| Pain scores for chest pain | x | x | x | x |
| CCQ | x | x | x | x |
| CAT | x | x | x | x |
| HADS | x | x | x | x |
| WPAI questionnaire | x | x | x | x |

CAT, COPD Assessment Test; CCQ, Clinical COPD Questionnaire score; ESWT, Endurance Shuttle Walk Test; HADS, Hospital Anxiety and Depression Scale; ISWT, Incremental Shuttle Walk Test; MRC, Medical Research Council; WPAI, Work Productivity and Activity Impairment Questionnaire.

## Primary outcome

The primary outcome of exercise capacity will be measured using the ISWT.[21] The ISWT will require patients to walk up and down a 10-metre course marked out using cones at walking speeds dictated by an audio signal played on an audio device. Each participant will receive standardised instructions to: 'Walk at a steady pace, aiming to turn around when you hear the signal. You should continue to walk until you feel that you are unable to maintain the required speed without becoming unduly breathless'. To account for any learning effect, a practice ISWT will be performed and the greatest distance walked from either test will be taken forward. The test will be terminated when either: (1) the patient indicates that they are unable to continue, (2) the operator determines that the patient is not fit to continue or (3) the operator assesses that the patient was unable to sustain the speed and cover the distance to the cone prior to the beep sounding.

## Secondary outcomes

### Exercise capacity and muscular strength

Participants will be asked to complete a five repetition sit to stand test,[24] as an indication of lower extremity strength. The time taken for participants to complete this task will be recorded.

The endurance shuttle walk test is a constant load walking test, in which a participant is required to walk at 85% of their maximal ISWT walking speed. An audio signal is played to the participant and they are required to walk around a 10-metre course marked by cones.[22]

### Symptoms

The MRC Dyspnoea Scale will be used to assess dyspnoea.[25] This is a 5-point questionnaire, in which the participant self-reports their dyspnoea. Chest pain will be assessed using four questions based on Brief Pain Inventory (Short Form).[26]

### Quality of life

The Clinical COPD Questionnaire[27] is a 10-item health-related quality of life questionnaire that is divided into three domains: symptoms, functional and mental. The COPD assessment test (CAT)[28] is a self-administered questionnaire that is used to quantify the symptom burden of COPD. The questionnaire consists of eight items, each with a 6-point scale, creating a total score out of 40. The EQ5D5L[29] will be used to assess generic health-related quality of life and comprises of five dimensions: mobility, self-care, usual activities, pain/discomfort and anxiety/depression. The questionnaire also contains a visual analogue scale, where the participant identifies their

health from 'the best you can imagine' and the 'worst you can imagine'.

### Psychological well-being

The self-reported Hospital Anxiety and Depression Scale[30] will be used to determine levels of anxiety and depression. The questionnaire consists of 14 questions with a 4-point Likert scale, and two 7-item subscales for anxiety and depression.

### Productivity

The Work Productivity and Activity Impairment questionnaire[31] will be used to assess productivity and impairments in paid/unpaid work and activities. The questionnaire is self-reported and participants are asked to recall the time missed from work in the last 7 days.

### Physical activity

Physical activity will be assessed using an ActiGraph wGT3X-BT activity monitor (ActiGraph, Pensacola, Florida). Participants will be asked to place the device on their right anterior hip during waking hours for 1 week prior to attending PR (baseline) and for 1 week prior to their 6-week follow-up visit. Participants will be asked to remove the device for water-based activities and for sleep. Written instructions will be provided to the participants. Full methodology for the assessment of physical activity is provided in online supplemental table 2.

### Cost/benefit analysis

The cost of the PR programme will be calculated, including single and recurrent costs. Single costs will include the necessary costs to set up and run PR, and the recurrent costs will include any item with a life expectancy of no more than 1 year, for example, disposable materials. Fixed costs will be captured before the first patient being recruited into the trial. Recurrent costs will be collected at the halfway stage of recruitment. An average cost of PR per patient will be calculated.

### Data management

Data collected during the study will be entered onto a database using Research Electronic Data Capture,[32 33] hosted by University of Leicester, UK. Access to the database will be via a secure password-protected web interface. Data will be validated using real-time data entry validation and electronic checks, led by the Independent Data Monitoring Committee (IDMC) at the University of Leicester, UK. The participants will be allocated a unique study-specific identification code.

### Data analysis

For stages 1 and 2b, qualitative data will be transcribed verbatim in Russian and translated into English for triangulation with UK-based researchers. Transcripts will be analysed using thematic analysis; following the six distinct stages of familiarisation with data; generating initial codes; searching for themes; reviewing themes; defining and naming themes and producing the report.[34 35] The qualitative lead will carry out initial coding and a sample of the transcripts will be coded independently by a second researcher to improve consistency and interpretive authenticity. The team will meet regularly to review emerging themes, with close attention to interactions within the interview and focus group data.

For stage 2a, quantitative analysis will be conducted using Statistical Package for the Social Sciences. Data will be reported as mean (SD), median (IQR) or frequency (%) as appropriate. All randomised patients will be included in an intention to treat analysis with the primary efficacy analysis based on both per protocol and modified intention-to-treat populations, with missing data imputed. For the primary analysis, the differences in the primary outcome of walking distance on the ISWT will be estimated using mixed models (unadjusted and adjusted for covariates if there are differences between groups at baseline). There will be no formal interim analysis of the data.

### Adverse events

All adverse events and serious adverse events will be recorded on an adverse event log. This will be recorded within the study trial management paperwork, CRF and study database. The IDMC will review high-level safety data, which will be monitored at least every month and on an ad hoc basis as required. The NIHR RECHARGE Scientific Committee will be informed of all adverse events and determine the need to terminate the trial prematurely. Participants who experience any such event will receive the appropriate care.

### Patient and public involvement

We know from conversations with adults living with PTBLD that support after completing TB treatment is lacking and that these individuals continue to live with a reduced quality of life, including respiratory symptoms and social isolation. It is clear that these individuals would benefit from support to better manage their condition, which is what this project seeks to deliver. The PR intervention and trial are to be informed directly from people living with PTBLD and healthcare workers in Kyrgyzstan. The results of this work will be presented to patients and the public at dedicated dissemination events, including those hosted at the NCCIM, Bishkek and surrounding TB Hospitals.

### Ethics and dissemination
#### Research ethics approval

The study was approved 22 July 2019 by Ethics Committee of the NCCIM (reference number 17) 3, Togolok Moldo Street, Bishkek, Kyrgyzstan. Ethics approval was also provided by the University of Leicester ethics committee on the 16 September 2019 ((reference number 22293, ethicsapp@leicester.ac.uk). Information about the study participants will be strictly confidential, and the names will be used only for internal reporting. No identifiable data will be published in any article.

## Dissemination policy

Study results will be disseminated through appropriate peer-reviewed journals, national and international conferences and through social media. All participants will be provided a summary of the trial results. In February 2021, a National Scientific Medical Forum is planned and a National Respiratory Congress is planned in October 2021, which will also present the results of the study and disseminate information about the project. In addition, mobile rehabilitation schools are planned during 2021.

**Author affiliations**
[1]Department of Pulmonology, National Center of Cardiology and Internal Medicine named after Academician M. Mirrakhimov, Bishkek, Kyrgyzstan
[2]Republican Research Center of Pulmonology and Rehabilitation, Bishkek, Kyrgyzstan
[3]Department of Respiratory Sciences, University of Leicester, Leicester, Leicestershire, UK
[4]Centre for Exercise and Rehabilitation Science, NIHR Leicester Biomedical Research Centre-Respiratory, University Hospitals of Leicester NHS Trust, Leicester, UK
[5]Faculty of Health and Human Sciences, University of Plymouth, Plymouth, UK
[6]South West Research Design Service, Plymouth, UK
[7]NIHR Leicester Biomedical Research Centre, University Hospitals of Leicester NHS Trust, Leicester, UK
[8]Department of Economics, University of Sheffield, Sheffield, UK
[9]University of Loughborough, Loughborough, UK

**Contributors** AA is the main author. MWO—Help in all sections of writing the Study Protocol. AVJ—Work on the structure of the manuscript and its skeleton. MM—Work on qualitative part of the Study Protocol. AM—Worked on the data analysis section. SA—Work on REDCap database adaptation. GM—Assistance in including the qualitative part in the protocol. AO—Help in developing a structure of pulmonary rehabilitation programme. KM—Help in developing an exercise and educational programmes. AT—Help in developing an educational programme. ARK—Help in developing an exercise programme. ZKY—Editorial work and a lot of work on the description of qualitative study. RJ—The inspirer of new ideas and unconventional approaches. AB- Editorial work. RM- Technical support and editorial work. AM—Database and storage organisation. JM—Technical support and editorial work. DM—Work on edition of qualitative section. RCF—Technical support and editorial work. MCS—Author of the main conceptual ideas. TS—Author of the idea and mastermind of the research protocol. SJS—Coauthor of the main idea.

**Funding** This research was funded by the National Institute for Health Research (NIHR) (17/63/20) using UK aid from the UK Government to support global health research. The views expressed in this publication are those of the author(s) and not necessarily those of the NIHR or the UK Department of Health and Social Care.

**Competing interests** None declared.

**Patient consent for publication** Not applicable.

**Provenance and peer review** Not commissioned; externally peer reviewed.

**Data availability statement** All authors have substantially contributed to the conception and design of the study. AA drafted the manuscript. All authors of the paper have revised the content and approved the final version to be published. All authors are accountable for all aspects of the work.

**ORCID iDs**
Azamat Akylbekov http://orcid.org/0000-0001-5761-399X
Mark W Orme http://orcid.org/0000-0003-4678-6574
Amy V Jones http://orcid.org/0000-0001-6565-8645
Maamed Mademilov http://orcid.org/0000-0001-8528-3115
Zainab K Yusuf http://orcid.org/0000-0001-7859-5102

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
