## [Reviewer comments · BMJ Open]

ARTICLE DETAILS

TITLE (PROVISIONAL)	Culturally adapted Pulmonary Rehabilitation for adults living with post-tuberculosis lung disease in Kyrgyzstan: Protocol for a randomised controlled trial with blinded outcome measures
AUTHORS	Akylbekov, Azamat; Orme, Mark; Jones, Amy; Mademilov, Maamed; Muratbekova, Aibermet; Aidaraliev, Shoir; Mirzaliev, Gulzada; Oleinik, Alena; Magdieva, Kamila; Taalaibekova, Aijan; Rysbek kyzy, Aidai; Yusuf, Zainab; Rupert, Jones; Barton, Andy; Miah, Ruhme; Manise, Adrian; Matheson, Jesse; Malcolm, Dominic; Free, Robert; Steiner, Michael; Sooronbaev, Talant; Singh, Sally

VERSION 1 – REVIEW

REVIEWER	Allwood, Brian Stellenbosch University
REVIEW RETURNED	23-Mar-2021

GENERAL COMMENTS	Thank you for the submission of this research protocol for submission. This protocol is a single centre, single blinded randomized control trial of pulmonary rehabilitation in patients with PTLT, with pre-trial qualitative research into the contextualization and review of the PR programme in Kyrgyzstan. Major Comment: PTLT is insufficiently defined, and a complicated issue. Please consult the recent Post-TB Symposium Proceedings (IJTLD 2020) for a more robust definition, or propose an alternative. Page 8 Line 11 – “any minor modifications” – It appears from this statement combined with Supp Table 1, that only minor modifications to the PR programme will be allowed after discussions in Stage 1. Please defend. What will happen if Stage 1 highlights major modifications that are needed? Page 8 Line 24 – An ineligibility rate of 70% is high, and will impact external validity of this study. Please explain how this number was derived at, and what measures will be put in place to improve external validity. Please describe more clearly, where patients will be recruited from (e.g. in- vs out-patient settings), and how patients will be approached for study participation (e.g. random, consecutive, or “my favourite patient” referrals). This is an important step in a single centre study, and if not well considered/documentated, will hamper the external validity and acceptance of the patient population/study results in the broader post-TB community.
---

	Minor Comment: Pg 5 Line 8 – “will be important for assessing impact of pulmonary rehabilitation in Kyrgyzstan, Central Asia and low- and middle-income countries (LMIC) more broadly, findings may not be generalizable to other parts of the world” – Please alter. Implications of the sentence as it stands is that all HICs are inherently different to Krygzstan, but all LMICs are the same as Krygzstan. I think this is a very narrow view, as there is huge heterogeneity in LMICs, for example, can the population of Krygzstan be compared to sub-Saharan Africa, central America and the Philippines? Highly unlikely. Pg 6 Line 35 – Please provide some examples of tests that the investigators will attempt to “contextualise”, depending on outcome of Stage 1, for the reader to better understand the extent of the PR programme modifications that the study will be attempting to implement. From this manuscript, it seems more likely that the changes will be around education of participants in how/why to do the tests prescribed, than altering the tests to there preferences/needs. Pg 7 line 17 – “Reasons for declining to participate will be reflected in the field notes.” – Duplication. Erase. Pg 8 Line 11 – “and for understand their views on any specific topics” – please rephrase.
--	---

REVIEWER	Balwan, Wahied Govt. Postgraduate College Bharderwah
REVIEW RETURNED	29-Apr-2021

GENERAL COMMENTS	The Manuscript has been well prepared. The Research article up to mark for getting published in Peer Reviewed journal.
--

REVIEWER	Schoch, Otto Kantonsspital St.Gallen, Pneumology and Centre for Sleep Medicine
REVIEW RETURNED	17-Sep-2021

GENERAL COMMENTS	Dear editors, dear authors Thank you for the opportunity to review this protocol and assess it for possible publication in BMJ open. The protocol deals with a very important aspect of tuberculosis, ie management and treatment of patients affected by post-tuberculosis lung disease. The treatment evaluated here is pulmonary rehabilitation and the data analysis focuses on a comparison between 2 randomized groups, those undergoing direct rehabilitation versus those with a start of rehabilitation delayed for at least 12 weeks, with random group allocation, blinded to those collecting data at 6 and 12 weeks. I think the study is very interesting and should be supported. However, I have some concerns, which should be clarified before publication of the protocol in BMJ open. Most of my points deal with the characteristics of the patient population, which are patients after tuberculosis treatment. I would thus propose to include more
---

precise information on the tuberculosis history, information which possibly might in future allow to target those patients after tuberculosis treatment who profit most.

Major suggestion / criticism:

1) Page 6, Line 56: Inclusion criteria in terms of TB status are not clear enough, some are actually very unclear, eg: «confirmed diagnosis of TB-negative patients with PTBLD... you mean sputum smear negative for acid fast bacilli will be included? Or smear negative as well as smear positive patients after successful TBTx?

2) I assume that patients who are still on TB medication will not be eligible for randomization, however, this is not stated explicitly.

3) Page 7, Line 5: evidence of active TB on Chest X-ray or sputum within 1 month of assessment: how would you define this? All participants must have 2 Chest X-rays within 1 month? (this is a bit unclear and quite critical) Suggest to use clinical parameters, weight etc to exclude active TB or recurrence and to classify TB treatment outcome categories according to WHO (see below).

4) Page 9, line 7f: Information on tuberculosis and it's treatment must be included in the baseline characteristics, for example timepoint of TB diagnosis (year, month), sputum smear status at TB diagnosis, duration of disease before TB treatment start (diagnostic delay), type and duration of TB Tx, time elapsed after TB treatment end.

In conclusion, I suggest collecting much more data on tuberculosis history and treatment and to define a minimum time lag between the end of TB treatment and the inclusion in the rehab program for PTBLD rehab. Best would be 6 months after the completion of TB-Treatment. This will allow to exclude early relapse cases and to define all patients as "Sustained treatment success" (for reference of recently updated WHO TB outcome definitions see ERJ 021 58: 2100804, DOI: 10.1183/13993003.00804-2021)

Minor points:

Page 8, line 52: "Recordings will be transcribed verbatim, with identifiable information removed." Should most likely read .. with NO identifiable information removed."

Page 9, line 24: Primary outcome measure is ISWT – secondary outcome measure is ESWT. Why did you choose like that? The Sit-to stand test is not mentioned here.

Page 11, line 39f: The dates to distribute results must be adapted, they are in the past.

VERSION 1 – AUTHOR RESPONSE

Reviewer: 1

Dr. Brian Allwood, Stellenbosch University

Comments to the Author:

Thank you for the submission of this research protocol for submission. This protocol is a single centre, single blinded randomized control trial of pulmonary rehabilitation in patients with PTLT, with pre-trial qualitative research into the contextualization and review of the PR programme in Kyrgystan.

Major Comment:

PTLD is insufficiently defined, and a complicated issue. Please consult the recent Post-TB Symposium Proceedings (IJTLD 2020) for a more robust definition, or propose an alternative.

Thanks a lot for your helpful comment. I listened to it and changed the definition according to Post-TB Symposium Proceedings (IJTLD 2020), and I admit that this definition better describes this condition.

Page 8 Line 11 – “any minor modifications” – It appears from this statement combined with Supp Table 1, that only minor modifications to the PR programme will be allowed after discussions in Stage 1. Please defend. What will happen if Stage 1 highlights major modifications that are needed?

Thank you for your comment. Sorry, but this expression should have read "even any minor modifications will be taken into account." We have now changed the wording to "any modifications". On the basis of qualitative research conducted, in truth, there have been some major modifications to the programme, e.g., brand new elements of the exercise and education components will be included in our PR intervention.

Page 8 Line 24 – An ineligibility rate of 70% is high, and will impact external validity of this study. Please explain how this number was derived at, and what measures will be put in place to improve external validity.

Thank you for raising this. This estimate was taken from a previous PR study conducted in Uganda. Therefore, we agree that it may not be suitable for our study. Therefore, we have removed this estimate from the manuscript and will report screening data in the results paper in accordance with CONSORT. We have amended the text to ensure clarity of our sample size calculation.

Please describe more clearly, where patients will be recruited from (e.g. in- vs out-patient settings), and how patients will be approached for study participation (e.g. random, consecutive, or “my favourite patient” referrals). This is an important step in a single centre study, and if not well considered/documentated, will hamper the external validity and acceptance of the patient population/study results in the broader post-TB community.

This is a very important point indeed. Thanks for pointing this out. I have provided details on this procedure in paper [OMW1][Y3M2] and will quote here:

In the databases of the above medical institutions there is information about patients who completed anti-tuberculosis treatment and clinically and laboratory identified as a cured (TB negative) patient. Such patients usually come for routine or follow-up consultations and their attending physicians can refer them to the PR study. Family doctors in charge of individual districts usually have cards of their patients with their diagnoses, analyses of their condition during the period of their supervision, as well as their contact information. In the case of family physicians, they can call patients with a history of

pulmonary tuberculosis who have completed treatment, are laboratory negative but still have respiratory symptoms, and refer them to PR.

Minor Comment:

Pg 5 Line 8 – “will be important for assessing impact of pulmonary rehabilitation in Kyrgyzstan, Central Asia and low- and middle-income countries (LMIC) more broadly, findings may not be generalizable to other parts of the world” – Please alter. Implications of the sentence as it stands is that all HICs are inherently different to Krygzstan, but all LMICs are the same as Krygzstan. I think this is a very narrow view, as there is huge heterogeneity in LMICs, for example, can the population of Krygzstan be compared to sub-Saharan Africa, central America and the Philippines? Highly unlikely.

You are correct, not all low- and middle-income countries are similar in context. Kyrgyzstan is more similar to other Central Asian countries and some other countries of the former Soviet Union. I have made some corrections to this section accordingly. I will bring the changes here:

Findings will be important for assessing impact of PR in Kyrgyzstan, as well as for Central Asian countries more broadly. By nature of developing culturally appropriate pulmonary rehabilitation, findings are unlikely to be directly transferrable to other low- and middle-income countries (LMICs), but may act as a model for developing local programmes globally.

Pg 6 Line 35 – Please provide some examples of tests that the investigators will attempt to “contextualise”, depending on outcome of Stage 1, for the reader to better understand the extent of the PR programme modifications that the study will be attempting to implement. From this manuscript, it seems more likely that the changes will be around education of participants in how/why to do the tests prescribed, than altering the tests to there preferences/needs.

Thank you for pointing this out. This will really help the reader to understand it more deeply. I have tried to provide some examples of what changes may be requested and needs, which are given below:

Focus groups with adults living with PTBLD and interviews with staff who would be involved in referring patients to PR will be conducted to explore initial perceptions of PR and to inform the cultural adaptation of PR, including any modifications (e.g. additional types of exercises, suggestions for the timing of classes, priorities for education content) and any additional needs (e.g. transport requirements).

Pg 7 line 17 – “Reasons for declining to participate will be reflected in the field notes.” – Duplication. Erase.

Thanks a lot. Mechanical error, my oversight. Fixed.

Pg 8 Line 11 – “and for understand their views on any specific topics” – please rephrase.

Thanks. I have rephrased the whole paragraph I quoted above:

Focus groups with adults living with PTBLD and interviews with staff who would be involved in referring patients to PR will be conducted to explore initial perceptions of PR and to inform the cultural adaptation of PR, including any modifications (e.g. additional types of exercises, suggestions for the timing of classes, priorities for education content) and any additional needs (e.g. transport

requirements).

Reviewer: 2

Dr. Wahied Balwan, Govt. Postgraduate College Bheaderwah

Comments to the Author:

The Manuscript has been well prepared. The Research article up to mark for getting published in Peer Reviewed journal.

Thank you very much. It is really encouraging and supportive.

Reviewer: 3

Dr. Otto Schoch, Kantonsspital St.Gallen

Comments to the Author:

Dear editors, dear authors

Thank you for the opportunity to review this protocol and assess it for possible publication in BMJ open.

The protocol deals with a very important aspect of tuberculosis, ie management and treatment of patients affected by post-tuberculosis lung disease. The treatment evaluated here is pulmonary rehabilitation and the data analysis focuses on a comparison between 2 randomized groups, those undergoing direct rehabilitation versus those with a start of rehabilitation delayed for at least 12 weeks, with random group allocation, blinded to those collecting data at 6 and 12 weeks. I think the study is very interesting and should be supported. However, I have some concerns, which should be clarified before publication of the protocol in BMJ open. Most of my points deal with the characteristics of the patient population, which are patients after tuberculosis treatment. I would thus propose to include more precise information on the tuberculosis history, information which possibly might in future allow to target those patients after tuberculosis treatment who profit most.

Major suggestion / criticism:

1) Page 6, Line 56: Inclusion criteria in terms of TB status are not clear enough, some are actually very unclear, eg: «confirmed diagnosis of TB-negative patients with PTBLD... you mean sputum smear negative for acid fast bacilli will be included? Or smear negative as well as smear positive patients after successful TBTx?

Thank you very much for this comment. I will try to clarify. Patients who have had tuberculosis for a long time are undergoing medical examination at the Bishkek Tuberculosis Dispensary under long-term supervision. Patients finish therapy, the TB specialist [OMW3][Y3M4] issues a document confirming recovery of the patient and diagnoses of PTBLD. After that, these patients are transferred to their family doctors for supervision.

Family doctors at their sites are supervising several families, including patients with PTBLD. That is, patients enrolled in the study have completed therapy and are under the supervision of their family doctors or a phthisiatrician from a TB dispensary. They come with a document from the phthisiatrician about recovery and are supported by a sputum analysis, which is considered only for greater safety when visiting the hospital where the study is taking place.

I took into account your comment and expanded these requirements as follows:

For Stage 1, eligible patients will be: aged ≥ 18 years; confirmed diagnosis of a TB-negative patients with PTBLD using a Ziehl-Nielsen stain or GeneExpert method and completed TB treatment. **Eligible HCPs will be directly involved in the long-term care of PTBLD patients.**

2) I assume that patients who are still on TB medication will not be eligible for randomization, however, this is not stated explicitly.

Thank you for your comment. Yes it is. Patients receiving anti-tuberculosis treatment in our health care system are diagnosed with active tuberculosis process and receive either inpatient or outpatient treatment, such patients are not suitable. Our referrers are their supervisors, so they will not send them for the Study.

3) Page 7, Line 5: evidence of active TB on Chest X-ray or sputum within 1 month of assessment: how would you define this? All participants must have 2 Chest X-rays within 1 month? (this is a bit unclear and quite critical) Suggest to use clinical parameters, weight etc to exclude active TB or recurrence and to classify TB treatment outcome categories according to WHO (see below).

Thanks for this comment, I think it's important too. Only 1 x-ray or 1 sputum test is enough and patients will not be eligible if there is evidence of active TB on Chest X-ray or sputum tests within 1 month of their initial pulmonary rehabilitation assessment. This is insurance as we need to be safe. As I said above, patients are under the supervision of their family doctors or phthisiatricians from the tuberculosis dispensary, who are constantly in touch with their patients.

4) Page 9, line 7f: Information on tuberculosis and it's treatment must be included in the baseline characteristics, for example timepoint of TB diagnosis (year, month), sputum smear status at TB diagnosis, duration of disease before TB treatment start (diagnostic delay), type and duration of TB Tx, time elapsed after TB treatment end.

Thank you for this comment and suggestion. This is certainly great and could very well cover the TB situation in Kyrgyzstan, but we left the diagnosis of the disease for specialists in this area. It so happened that respiratory medicine and phthisiology are separate disciplines in our country and in the post-Soviet context. We work with patients with residual lung changes and, accordingly, some respiratory symptoms (primarily shortness of breath) who have already been diagnosed with PTBLD.

In conclusion, I suggest collecting much more data on tuberculosis history and treatment and to define a minimum time lag between the end of TB treatment and the inclusion in the rehab program for PTBLD rehab. Best would be 6 months after the completion of TB-Treatment. This will allow to exclude early relapse cases and to define all patients as "Sustained treatment success" (for reference of recently updated WHO TB outcome definitions see ERJ 021 58: 2100804, DOI: 10.1183/13993003.00804-2021)

Thank you for your comment. I listened to your advice. 6 months is a really good threshold.

Minor points:

Page 8, line 52: "Recordings will be transcribed verbatim, with identifiable information removed." Should most likely read .. with NO identifiable information removed."

Thank you for your comment. This means that their personal information (name, age, place of residence) is removed.

Page 9, line 24: Primary outcome measure is ISWT – secondary outcome measure is ESWT. Why did

you choose like that? The Sit-to stand test is not mentioned here.

We are happy to clarify this decision. The ISWT is sensitive to change and is much more commonly collected compared with the ESWT. The ISWT also forms part of our Global RECHARGE minimum dataset (available here: <https://www.ncbi.nlm.nih.gov/pmc/articles/PMC7688060/>) and so ensuring comparability across our sites and future work in this area. The 5x sit to stand is mentioned under the 'Exercise capacity and muscular strength' subheading.

Page 11, line 39f: The dates to distribute results must be adapted, they are in the past.

Thank you for highlighting these errors. The text has now been amended accordingly.

Reviewer: 1

Competing interests of Reviewer: None.

Reviewer: 2

Competing interests of Reviewer: Article is very well written and apt to be published.

Reviewer: 3

Competing interests of Reviewer: no

VERSION 2 – REVIEW

REVIEWER	Allwood, Brian Stellenbosch University
REVIEW RETURNED	07-Dec-2021

GENERAL COMMENTS	Manuscript Title: "Culturally adapted Pulmonary Rehabilitation for adults living with post-tuberculosis lung disease in Kyrgyzstan: Protocol for a randomised controlled trial with blinded outcome measures." I am happy that the authors have adequately addressed concerns around the first submission of this manuscript, which is a detailed description of an RCT protocol for pulmonary rehabilitation in PTLTD.
--

REVIEWER	Schoch, Otto Kantonsspital St.Gallen, Pneumology and Centre for Sleep Medicine
REVIEW RETURNED	10-Dec-2021

GENERAL COMMENTS	All my concerns and comments have been addressed in the revised manuscript, thank you
---